# Investigation on the Coupling Effects between Flow and Fibers on Fiber-Reinforced Plastic (FRP) Injection Parts

**DOI:** 10.3390/polym12102274

**Published:** 2020-10-03

**Authors:** Chao-Tsai Huang, Cheng-Hong Lai

**Affiliations:** Department of Chemical and Materials Engineering, Tamkang University, No. 151, Yingzhuan Rd., Tamsui Dist., New Taipei City 25137, Taiwan; leo0988813302@gmail.com

**Keywords:** injection molding, fiber orientation distribution, flow–fiber coupling, fiber reinforced plastics (FRP)

## Abstract

Glass or carbon fibers have been verified that can enhance the mechanical properties of the polymeric composite injection molding parts due to their orientation distribution. However, the interaction between flow and fiber is still not fully understood yet, especially for the flow–fiber coupling effect. In this study, we have tried to investigate the flow–fiber coupling effect on fiber reinforced plastics (FRP) injection parts utilizing a more complicated geometry system with three ASTM D638 specimens. The study methods include both numerical simulation and experimental observation. Results showed that in the presence of flow–fiber coupling effect, the melt flow front advancement presents some variation, specifically the “convex-flat-flat” pattern will change to a “convex-flat-concave” pattern. Furthermore, through the fiber orientation distribution (FOD) study, the flow–fiber coupling effect is not significant at the near gate region (RG). It might result from the strong shear force to repress the appearance of the flow–fiber interaction. However, at the end of filling region (ER), the flow–fiber coupling effect tries to diminish the flow direction orientation tensor component A_11_ and enhance the cross-flow orientation tensor component A_22_ simultaneously. It results in the dominance in the cross-flow direction at the ER. This orientation distribution behavior variation has been verified using a micro-computerized tomography (micro-CT) scan and image analysis technology.

## 1. Introduction

As the human population keeps increasing dramatically, CO_2_ emissions from transportation based on the burning of fossil fuels will continue to be a major contributor to air pollution [1,2,3,4]. To decrease the CO_2_ emissions from transportation, one of the most effective methods is using lightweight materials, such as fiber-reinforced plastics (FRP), to enhance the fuel efficiency [5,6,7]. To understand how fibers can enhance the mechanical strength of FRP products, many researchers have tried to study their features experimentally [8,9,10,11]. However, there are at least two major challenges which people still encounter in FRP injection molding. Specifically, one is what the fiber features are under the influence of the polymer matrix flow dynamic, and the other is how the fiber will affect the polymer matrix flow dynamic and generate the coupling between polymer flow and fibers.

One of the key fiber features in FRP injection molding is the flow-induced fiber orientation behavior. Folger and Tucker [12], Advani and Tucker [13], and Advani [14] have proposed some numerical models to predict fiber orientation in short fiber reinforced plastics injection molding. These models have given great guidance for people to understand fiber orientation behavior and develop good FRP products [15,16,17]. However, when the fiber length is increased in the FRP injection molding, the physical behaviors of the molten plastics become much more complicated. To discover the microstructures of longer fiber reinforced plastics, many researchers have developed several theoretical models. Phelps and Tucker [18] proposed the Anisotropic Rotary Diffusion (ARD) model for short to long fibers. Wang and Tucker [19] developed the Reduced Strain Closure (RSC) model. Then the ARD and RSC have been integrated into the ARD–RSC model and implemented into a commercial injection molding simulation software, Autodesk Moldflow Insight (AMI) [18,20]. Specifically, there are six parameters and inlet condition needs to be specified for the ARD–RSC model. It is very complicated to obtain suitable parameters to avoid numerical instability for numerical computation. To improve the above challenges, Tseng et al. [21,22,23,24] have developed the improved ARD model and Retarding Principal Rate (known as the iARD–RPR) model. Based on the iARD–RPR model, people only need to specify three physical meaning parameters: a fiber–fiber interaction parameter, a fiber–matrix interaction parameter, and a slow-down parameter. The iARD–RPR model has been adopted in a commercial injection molding software, Moldex3D.

Furthermore, the fibers and their orientations will also influence the flow field as well. In fact, many researchers have been studied the flow–fiber coupling effect. Lipscomb et al. [25] developed a constitutive equation of the extra stress tensor to consider the fiber orientation effect on flow stress. Later, Ver Weyst and Tucker [26] developed a finite element method to cover flow–fiber coupled problems and applied to some geometrical systems based on the Hele–Shaw model. Specifically, at a center-gated disk system, under the fiber effects on the flow field, the velocity profile did not change significantly. In addition, when melt goes far from the gate, they found out that flow–fiber coupling is not significant. Moreover, to understand the fiber–flow effect in 3D flow system, Tseng and Su [27] have implemented the Lipscomb constitutive equation in Moldex3D software. They found out the velocity profile almost does not change with respect to Np (a dimensionless parameter that characterizes the effect of the fibers on the viscosity). The overall flow–fiber coupling effect by using 3D numerical calculation is similar to that by using the Hele–Shaw approximation of Ver Weyst and Tucker [26]. However, when they considered higher flow–fiber coupling effects, the 3D numerical calculation became unstable. To solve this numerical difficulty, Favaloro et al. [28] developed a scalar viscosity model (so called informed isotropic, IISO, model) for fiber suspensions to enhance the calculation of the flow–fiber coupling effect. The IISO viscosity model can be used to predict the wider core orientation distribution region. Moreover, Tseng and Favaloro [29] have further proposed a more comprehensive IISO fiber suspension constitutive equation (called the revised IISO model) which includes the strain-rate dependent parameter of the Trouton ratio. They showed that using this model on the complex injection molding flow of FRP, a special “convex-flat-concave” flow pattern and a wider core fiber orientation distribution can be obtained in the end-gated plate system. However, the flow–fiber coupling effect is still not comprehensively understood for the practical industrial cases yet.

In this study, we have tried to understand how the flow–fiber coupling effect is in more practical cases. Specifically, the flow–fiber coupling effect on FRP injection parts is investigated using a geometry system with three ASTM D638 specimens. During this investigation, the numerical simulation based on the iARD–RPR model of Tseng et al. [21,22,23,24] and the revised IISO model of Tseng and Favaloro [29] is utilized to predict the fiber orientation distribution (FOD) either in the presence or in the absence of the flow–fiber coupling. Moreover, to validate the numerical prediction, the mechanical properties (tensile strength and tensile modulus) have been utilized to check the fiber microstructure and macro-property relationship followed the Ref. [30]. Furthermore, the FOD of real specimens have been studied by using micro-computerzied tomography (micro-CT) technology to scan the specimen. The scanned images were further analyzed using AVIZO^®^ software. Moreover, the comparison of numerical prediction and experimental observation of FOD was performed to validate the effect of flow–fiber coupling. To get a better understanding, the rest of this paper is organized as follows. In Section 2, the theoretical background is presented. Moreover, the injection molding system and related information have been addressed in Section 3. In addition, some experimental equipment and parameters are also described. In Section 4, the results and discussion will be addressed. Finally, a brief conclusion will be given in Section 5.

## 2. Theoretical Background

The theories regarding the FRP materials in the injection molding process will be divided into three parts: the polymer fluid mechanics, the fiber orientation kinetics, and the evolution of viscosity by flow–fiber coupling separately. The details are as follows.

### 2.1. Models for Polymer Fluid Mechanics

The polymeric melt is considered as compressible, non-Newtonian fluids. The governing equations for the polymer fluid mechanics, which are regarded as 3D transient non-isothermal motion, are:(1)$$\frac{\partial \rho }{\partial t}+\nabla \cdot \rho u=0$$
(2)$$\frac{\partial }{\partial t}\left(\rho u\right)+\nabla \cdot \left(\rho uu\right)=\nabla \cdot \sigma +\rho g$$
(3)σ=−PI+τ
(4)$$\rho C_{P}\left(\frac{\partial T}{\partial t}+u\cdot \nabla T\right)=\nabla \cdot \left(k\nabla T\right)+\tau :D$$
where ρ is density; **u** is velocity vector; t is time; **σ** is total stress tensor; **τ** is extra stress tensor; **g** is the acceleration vector of gravity; *P* is pressure; C_P_ is specific heat; T is temperature; *k* is thermal conductivity; **D** is the rate-of-deformation tensor.

For the polymer melt, the extra stress tensor without flow–fiber coupling can be expressed as:(5)τ=2ηD
where *η* is the shear viscosity of a polymer melt; it is a function of temperature and shear rate as described in Equations (6) and (7).

Moreover, one of the most important properties to influence the flow mechanics is viscosity, which is a strong function of temperature and shear rate. To estimate the shear rate and temperature dependence of viscosity, the viscosity characterization has been performed using a rheometer from CoreTech System (Moldex3D) Co. Ltd. Furthermore, the measured data has been executed with curve-fitting using various viscosity models. One of the best fitted is based on the modified-Cross model. Hence, the modified-Cross model with Arrhenius temperature dependence is employed to describe the viscosity of a polymer melt in this study:(6)$$\eta \left(T,\dot{\gamma }\right)=\frac{\eta_{o}\left(T\right)}{1+{\left(\eta_{o}\dot{\gamma }/\tau^{*}\right)}^{1-n}}$$
with
(7)$$\eta_{o}\left(T\right)=BExp\left(\frac{T_{b}}{T}\right)$$
where *n* is the power law index; *η_0_* is the zero shear viscosity; *τ** is the parameter that describes the transition region between zero shear rate and the power law region of the viscosity curve; *B* is constant; *T_b_* is a reference temperature.

### 2.2. Models for Fiber Orientation Kinetics

During the injection molding processing, the orientation of the fiber inside the FRP composite will be affected by the polymer fluid mechanics. In general, the orientation of a single fiber can be at an arbitrary direction, or at one of the three principal directions, as shown in Figure 1. However, since there are hundreds of thousands of fibers inside the polymer matrix in injection molding, it is too difficult to observe manually. Instead, to predict the fiber orientation kinetics, each single fiber is theoretically regarded as an axisymmetric bond with rigidness. The bond’s unit vector **p** along its axis direction can be described as the fiber orientation. The orientation state of a group of fibers is given by the second-order orientation tensor,
(8)A=∮ψ(p)pp dp
where ψ(p) is the probability density distribution function over the orientation space. Tensor **A**_4_ is a fourth order orientation tensor, defined as:(9)$$A_{4}=\oint \psi \left(p\right)pppp\;dp$$
where this tensor is also symmetric. Among the nine components of **A***_ij_*, only five are independent. Three of the remaining components are determined by symmetry:**A***_ij_* = **A***_ji_*(10)

The last component is determined by
**A***_11_* + **A***_22_* + **A***_33_* = 1(11)

The acceptable calculation is obtained through the eigenvalue-based optimal fitting approximation of the orthotropic closure family.

To handle this complicated tensor system, Tseng et al. [21,22,23,24] extended the ARD–RSC models [18,19,20] to develop a fiber orientation model to couple with Jeffery’s hydrodynamic (HD) model, namely, the iARD–RPR model (known as the Improved Anisotropic Rotary Diffusion model combined with the Retarding Principal Rate model),
(12)$$\dot{A}={\dot{A}}_{HD}+{\dot{A}}_{\mathrm{iARD}}\left(C_{I},\;C_{M}\right)+{\dot{A}}_{\mathrm{RPR}}\left(\alpha \right)$$
where $\dot{A}$ represents the material derivative of A. Parameters C_I_ and C_M_ describe the fiber–fiber interaction and fiber–matrix interaction, while parameter α can slow down the response of fiber orientation. The Jeffery’s hydrodynamic (HD) term can be written as
(13)$${\dot{A}}_{\mathrm{HD}}=\left(W\cdot A-A\cdot W\right)+\xi \left(D\cdot A+A\cdot D-2A_{4}:D\right)$$
(14)$$D\;=\;\frac{1}{2}\left(\nabla u+\nabla u^{T}\right)$$
(15)$$W\;=\;\frac{1}{2}\left(\nabla u-\nabla u^{T}\right)$$
where **W** and **D** are the vorticity tensor and the rate-of-deformation tensor, respectively. ξ is a shape factor of a particle. The rest of the details for the RPR model and the iARD model are available elsewhere [29].

### 2.3. Models for Evolution of Viscosity by Flow–Fiber Coupling

It was mentioned earlier that the fibers will affect the polymer fluid mechanics, and flow will further influence fibers back and forth; it introduces the coupling effect between the flow and fibers. To predict this flow–fiber coupling, the revised IISO constitutive equation has developed to handle the complex flow–fiber coupling by Tseng and Favaloro [29]. This revised IISO constitutive equation has been implemented into the commercial injection molding software, Moldex3D. During the injection molding simulation, the governing equations of flow field and fiber orientation are solved by the 3D finite volume method based on 3D geometry. After the velocity field and the fiber orientation filed are obtained at the beginning, they are used to determine the revised IISO viscosity. Then the iteration keeps going till they are converged. The details are available elsewhere [29]. Specifically, the revised IISO viscosity is presented as below.
(16)$$\eta^{IIOS}=\left(1+R_{T}K_{s\;}\right)\eta_{S}$$
(17)$$R_{T}(\dot{\dot{\gamma })=\frac{R_{T}^{0}}{1+{\left(\dot{\dot{\gamma }/{\dot{\gamma }}_{C}}\right)}^{2}}}$$
(18)$$K_{s}=\frac{D:A_{4}:D}{2D:D}$$
where $\eta_{S}$ the nonlinear Newtonian viscosity for the fiber-filled polymer fluids and is described by the modified-Cross model; *R_T_* is the dimensionless Trouton ratio parameter as a function of the strain rate; $R_{T}^{0}$ is the initial value of *R_T_*; *K_S_* is a stretching kernel which is related to the flow fields and to the fiber orientation state; ${\dot{\gamma }}_{C}$ is the critical strain rate (1/s).

## 3. Injection Molding System and Related Information

### 3.1. Simulation Model and Related Information

The geometrical model is shown as in Figure 2. Specifically, the dimension of the whole model is 400 mm × 165 mm × 3 mm. There are three standard specimens based on ASTM D638, marked as Model I, II, and III, respectively. The dimension for each standard specimen is 172 mm × 20 mm × 3 mm. To study the flow field behavior, we have designed those three models with different gate types. In particular, one has an edge gate, another has a sprue gate, and the other has a double gate, as presented. To fill the polymer melt into those three specimens, there is one plunger type melt entrance with 40 mm diameter that is located in the center of the cavity. Moreover, the moldbase and cooling channel layout are presented in Figure 3.

Furthermore, several measuring nodes in different models are specified to study the fiber orientation distribution (FOD), as in Figure 4. In particular, they are specified as three regions for observation for each model: the gate region (GR), the center region (CR), and the end of filling region (ER), respectively. Moreover, to perform the injection molding simulation, the associated process condition setting is as follows. Specifically, the filling time is 1.49 s, the packing time is 5 s, the cooling time is 15 s, and other parameters are in Table 1. The materials used in this study are pure polypropylene (later called PP) and polypropylene with short fiber of 3 mm length (later called PP + SF). Both PP and PP + SF materials are commercially available and supplied by LCY Chemical. Moreover, the associated material properties including viscosity, the specific volume against temperature and pressure (pvT), heat capacity (Cp), and thermal conductivity (K) have been measured and saved as material data by Moldex3D. In addition, to study the fiber orientation effect during the injection molding, the iARD–RPR model is adopted. In general, those three parameters are suggested in 0 < C_I_ < 1, 0 < C_M_ < 1, and 0 < α < 1 [29]. Here, the related parameters for the iARD–RPR model in this study are suggested from Moldex3D, as listed in Table 2. Furthermore, to conduct the flow–fiber effect, the revised IISO model is utilized. The associated parameters are also suggested from Moldex3D, which are presented in Table 3.

### 3.2. Experimental Equipment and Related Information

Figure 5a displays the injection machine (Chuan Lih Fa Machinery Works Co., Tainan, Taiwan Model: CLF-180TXL). Figure 5b shows the real mold structure and cooling channel layout. The PP and PP + SF materials are supplied by LCY Chemical as mentioned earlier. Specifically, the grade name: PP is Globalene ST868M and PP + SF is Globalene SF7351. Moreover, to perform the real injection molding testing, the associated process condition setting is the same as that of the simulation, as listed in Table 1. In order to validate the flow–fiber coupling effect calculated from numerical simulation, the mechanical properties via tensile test for the real injected parts have been executed using the universal tensile machine from Shimadzu, Kyoto, Japan (AGS-J mode). In addition, to validate the fiber orientation behavior, micro-computerized tomography (μ-CT) technology has been performed using Bruker Skyscan 2211 with 40–190 kV and a resolution of 5 μm, supported by Material and Chemical Laboratories (MCL) Multiscale X-ray CT laboratory, Industrial Technology Research Institute, Hsinchu, Taiwan.

## 4. Results and Discussion

To understand what the coupling effect is between flow and fiber in the injection molding process, we have tried to observe the mechanical property (macroscopic property) in the presence of fibers for FRP first. Furthermore, the fiber orientation behavior (microscopic feature) has been discovered. Through the relationship between mechanical properties and fiber orientation behavior, the flow–fiber coupling effect can be further discussed. The details are as follows.

### 4.1. Mechanical Property Test

To conduct the fiber reinforced capability, tensile testing has been performed using the universal tensile testing machine of Shimadzu (AGS-J mode). The testing method and procedures are based on the Ref. [31]. During the testing, the deformation and the force were recorded. The deformation and the force can be converted to the stress and strain using the following equations:(19)$$\sigma_{s}=\frac{F}{A_{0}}$$
(20)$$\varepsilon =\frac{L-L_{0}}{L_{0}}\times 100\%=\frac{\Delta L}{L_{0}}\times 100\%$$
(21)$$E=\frac{\sigma_{3.5\%}-\sigma_{2.5\%}}{\varepsilon_{3.5\%}-\varepsilon_{2.5\%}}$$
where σ_σ*σ*_ is tensile stress; F is the pulling force on the original cross-section area (A_0_) of the narrow portion of the standard specimen; ε is the elongation ratio; (L_0_) is the original length of the narrow portion; L is deformation; E is the tensile modulus.

For each tensile testing, those three models of ASTM D638 standard specimens have been installed into the tensile machine under a constant strain at 20 mm/min. For each model, five specimens have been executed for the same testing. After the five tests, the average stress–strain relation is presented in Figure 6. From this stress–strain relation in Figure 6a, the tensile modulus is calculated around the strain range of 2.5–3.5% to avoid the influence from the testing apparatus. The tensile modulus are obtained as 1463.67 N/mm^2^ for Model I, 1427.33 N/mm^2^ for Model II, 1299.25 N/mm^2^ for Model III, respectively. Based on the tensile modulus, Model I is stronger than Model II, while Model III is weakest due to the double gate design. Moreover, the tensile strength is also conducted for those three models as listed in Figure 6b. Additionally, the tensile strength for each testing has been recorded as in Table 4. Specifically, when material is changed from pure PP to PP + SF, the tensile strength of Model I is increased from 23.35 to 75.71 N/mm^2^. It is increased from 23.04 to 73.34 N/mm^2^ for Model II, and from 22.65 to 34.38 N/mm^2^ for Model III. Although the tensile modulus and tensile strength of the Model I and Model II are close, the actual measured data (including stress–strain curve, stress modulus, and tensile strength) of Model I are greater than that of Model II, as shown in Figure 6 and Table 4. Overall, the mechanical properties of Model I are stronger than those of Model II, while Model III’s are weakest due to the double gate design.

### 4.2. Fiber Orientation Effects With and Without Flow–Fiber Coupling

To realize why the mechanical properties of Model I are stronger than those of Model II and then to discover the flow–fiber coupling effect, the fiber orientation features have been utilized. The fiber orientation tensor components are predicted numerically by using Moldex3D R16^®^. They are also validated experimentally by using a micro-computerized tomography (micro-CT) scan of the specimens and utilizing Avizo^®^ for image analyses. The details are as follows. Figure 7a shows the fiber orientation distribution (FOD) predicted numerically without considering flow–fiber coupling at Point B for Model I. The flow direction orientation tensor component **A_11_** is increased from 0.7 to 0.78 at the skin-layer then decreased to 0.6 at the core-layer. Overall, the fiber orientation is dominant along the flow direction at gate region (GR). Moreover, Figure 7c shows the FOD observation experimentally. Except at little skin region, the flow direction orientation tensor component **A_11_** is around 0.8 to 0.9 at the skin-layer then decreased to 0.8 at the core-layer. It also shows that the dominant fiber orientation is at the flow direction in experiments. The simulation prediction of the FOD without considering flow–fiber coupling is in reasonable agreement with that of the experimental result for Model I at GR. Moreover, the simulation results of the fiber orientation tensor components are very close for both without and with flow–fiber coupling for Model I at GR, as shown in Figure 7a,b. Their trends are all consistent with the experimental observation. Since there is a strong entrance effect around GR [30], the fibers are influenced significantly by the higher shear rate (shear force) that will repress the flow–fiber coupling effect.

Moreover, considering the center region (CR), Figure 8a shows the FOD predicted numerically at Point E for Model I in the absence of flow–fiber coupling. The flow direction orientation tensor component **A_11_** is increased from 0.7 to 0.85 at the skin-layer then decreased to 0.75 at the core-layer. Overall, the fiber orientation is still dominant along the flow direction at CR. In addition, Figure 8c shows the FOD observation experimentally. The flow direction orientation tensor component **A_11_** is around 0.6 to 0.95 at the skin-layer then decreased to 0.85 at the core-layer. It also shows that the dominant fiber orientation is at the flow direction in experiments. Once again, the simulation prediction of the FOD in the absence of flow–fiber coupling is in a reasonable agreement with that of the experimental result for Model I at CR. Moreover, the simulation results of the fiber orientation tensor components for both without and with flow–fiber coupling for Model I at CR are very similar, as shown in Figure 8a,b. Due to the stronger shear rate (shear force) by the convergent section, the flow–fiber coupling is not significant at this region.

Furthermore, in the end of filling region (ER), Figure 9a shows the FOD predicted numerically without considering flow–fiber coupling at Point H for Model I. The flow direction orientation tensor component **A_11_** is increased from 0.65 to 0.85 at the skin-layer then decreased to 0.5 at the core-layer. Overall, the fiber orientation is still dominant along the flow direction at ER. Moreover, Figure 9c shows the FOD observation experimentally. The flow direction orientation tensor component **A_11_** is around 0.65 to 0.8 at the skin-layer then decreased dramatically to 0.4 at the core-layer. The fiber orientation is dominant at the flow direction **A_11_** at the skin-layer and then changes to the cross flow direction **A_22_** at the core-layer. From the experimental observation, the fiber orientation is strong in the cross-flow direction. It is noted that the simulation prediction without flow–fiber coupling over-predicted **A_11_** and under-predicted **A_22_** for Model I at ER. In addition, the width of the core-layer from the experiment is wider than that of the numerical prediction. Moreover, Figure 9b shows the simulation result under the influence of the flow–fiber coupling. Compared to the experimental observation, the simulation result with the flow–fiber coupling is much closer to the experimental observation. It is noted that during the melt flowing into the ER, the influence of the transport driving force is weaker and then the flow–fiber coupling becomes more significant. It could be the reason why the **A_11_** is over-predicted and **A_22_** is under-predicted by numerical simulation without considering the flow–fiber coupling effect.

Moreover, Figure 10a presents the FOD predicted numerically without consideration of flow–fiber coupling at Point B for Model II at GR. The flow direction orientation tensor component **A_11_** is 0.82 at the skin-layer then decreased to 0.4 at the core-layer. The fiber orientation is dominant at the flow direction **A_11_** at the skin-layer and then changes to the cross flow direction **A_22_** at the core-layer. Similarly, Figure 10c exhibits the FOD observation experimentally at Point B for Model II. It shows that the simulation prediction of the FOD without consideration of flow–fiber coupling is consistent with that of the experimental result for Model II at GR in the trend. However, the width of the core-layer by experimental observation is wider than that of simulation prediction. Furthermore, in Figure 10a,b, there is some difference on the simulation results of the fiber orientation tensor components between both without and with flow–fiber coupling cases for Model II at GR. Specifically, under the influence of flow–fiber coupling, more fibers will stay at the cross flow direction **A_22_** at the core-layer, as shown in Figure 10b. The trend of the coupled case is more consistent with the experimental observation, as shown in Figure 10c.

Furthermore, Figure 11a presents the FOD predicted numerically in absence of the flow–fiber coupling at Point E for Model II. The flow direction orientation of tensor component **A_11_** is 0.83 at the skin-layer then decreased to 0.75 at the core-layer. The fiber orientation is dominant at the flow direction **A_11_**. Additionally, Figure 11c exhibits the FOD observation experimentally at Point E for Model II. The flow direction orientation tensor component **A_11_** is 0.7 to 0.95 at the skin-layer, then decreases to 0.7 at the core-layer. Although the fiber orientation is still dominant at the flow direction **A_11_**, the fiber orientation prediction at the core-layer, predicted numerically without considering flow–fiber coupling, is over-predicted. Moreover, Figure 11a,b show the simulation results of the fiber orientation tensor components for both without and with flow–fiber coupling cases for Model II at CR. Due to the stronger shear rate (shear force) by the convergent section, flow–fiber coupling is not significant at this region except for the core-layer area. However, in the presence of flow–fiber coupling, the numerical prediction is closer to the experimental observation.

Moreover, Figure 12a shows the FOD predicted numerically without flow–fiber coupling at Point H for Model II. The flow direction orientation of tensor component **A_11_** is increased from 0.7 to 0.8 at the skin-layer, then decreased to 0.5 at the core-layer. Overall, the fiber orientation is still dominant along the flow direction at ER. In addition, Figure 12c shows the FOD observation experimentally. The flow direction orientation tensor component **A_11_** is around 0.6 to 0.8 at the skin-layer, then decreased dramatically to 0.3 at the core-layer. The fiber orientation is dominant at the flow direction **A_11_** at the skin-layer and then changes to the cross flow direction **A_22_** at the core-layer. From the experimental observation, the fiber orientation is strong in the cross-flow direction at the core-layer. The simulation prediction in the absence of flow–fiber coupling over-predicted **A_11_** and under-predicted **A_22_** for Model II at ER. Additionally, the width of the core-layer from the experiment is significantly wider than that of numerical prediction. Moreover, Figure 12b shows the fiber orientation tensor component at the ER for Model II under the influence of the flow–fiber coupling effect. Compared to the experimental observation, the simulation with flow–fiber coupling is much closer to the experimental result. It is noted that during the melt flowing into the ER, the influence of the transport driving force of Model II is much weaker and then the flow–fiber coupling becomes much more significant compared to Model I cases. Overall, from Model I and Model II, it can be concluded that when the melt flows through the higher shear rate (shear force) regions, such as GR and CR of Model I, the flow–fiber coupling will be repressed. However, when the shear rate (shear force) of the melt is reduced gradually (such as at the ER of Model I and at the CR and ER of Model II), the flow–fiber coupling effect will become more significant.

### 4.3. Correlation between Mechanical Property and Fiber Micro-Feature

Moreover, comparing the orientation tensor component **A_11_** of the experimental observation from GR via CR to ER for both Model I and Model II (from Figure 7c, Figure 8c, Figure 9c, Figure 10c, Figure 11c and Figure 12c), the **A_11_** component at the central line is increased by the convergent structure and then gradually reduced by the divergent structure. The variation of the FOD is quite different for Model I and II. Specifically, from Figure 7c and Figure 10c, the A_11_ of the Model I is much higher than that of Model II at GR. In addition, the **A_11_** of the Model I is also higher than that of Model II at CR and at ER. Overall, the stronger alignment in flow direction presented in **A_11_** of Model I is expected to provide stronger mechanical properties (including tensile modulus and tensile stress), which is consistent with that in mechanical property testing. The reason for this difference is due to the entrance effect that occurs when melt flows through the edge gate design. That entrance effect will further provide some flow-induced fiber orientation to the melt, which will further enhance the tensile properties of Model I, as described in Ref. [30]. Moreover, from the comparison of the fiber orientation tensor components between the numerical simulation and experimental results, the numerical simulation, with consideration of flow–fiber coupling, is much closer to the experimental observation.

### 4.4. Melt Flow Behavior under the Influence of Flow–Fiber Coupling

Finally, during the injection molding processing, the molten plastics are delivered from the sprue via the runner and gate and flow into the cavity. The melt flow front behavior is presented in Figure 13. Clearly, when the flow–fiber coupling is considered, the melt flow front exhibits “convex-flat-concave” behavior, while it shows “convex-flat-flat” behavior without flow–fiber coupling. This phenomenon is due to the free surface advancing faster along the side walls of the cavity due to the flow–fiber coupling effect. This flow–fiber coupling-induced melt flow behavior is similar to that described in Ref. [29].

## 5. Conclusions

In this study, we have tried to investigate the flow–fiber coupling effect on FRP injection parts using a more complicated geometry system with three ASTM D638 specimens through both simulation prediction and experimental observation. Results showed that in the presence of the flow–fiber coupling effect, the melt flow front advancement presents some variation, specifically the original “convex-flat-flat” pattern will change to a “convex-flat-concave” pattern. This observation is consistent with that of Tseng and Favaloro [30], in that it is a simple end-gated plate system. Furthermore, through the fiber orientation study, the flow–fiber coupling effect is not significant at the gate region (RG). It might result from the strong shear force to hold down the appearance of the flow–fiber coupling. However, at the end region (ER), since the shear force becomes lower, the flow–fiber coupling effect tries to diminish the flow direction orientation tensor component **A_11_** and enhance the cross-flow orientation tensor component **A_22_** simultaneously. It ends up with the cross-flow direction dominance at the ER. This fiber orientation distribution (FOD) behavior variation to cause the flow–fiber coupling behavior has been verified using micro-computerized tomography (μ-CT) scan and images analyses. This overall **A_11_** dominance of the FOD arrangement for Model I has provided stronger tensile properties that are consistent with the experimental observation via tensile test.

## Figures and Tables

**Figure 1 polymers-12-02274-f001:**
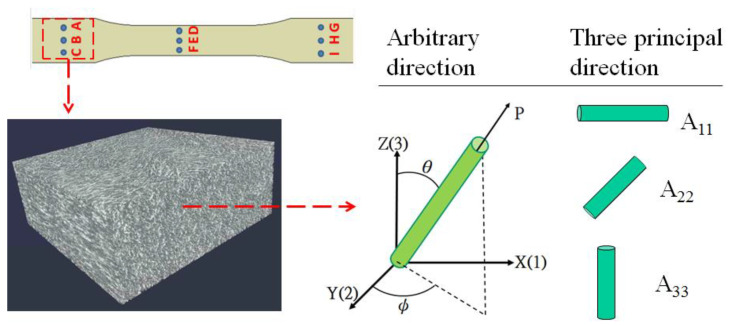
The schematic of the fiber orientation tensor components.

**Figure 2 polymers-12-02274-f002:**
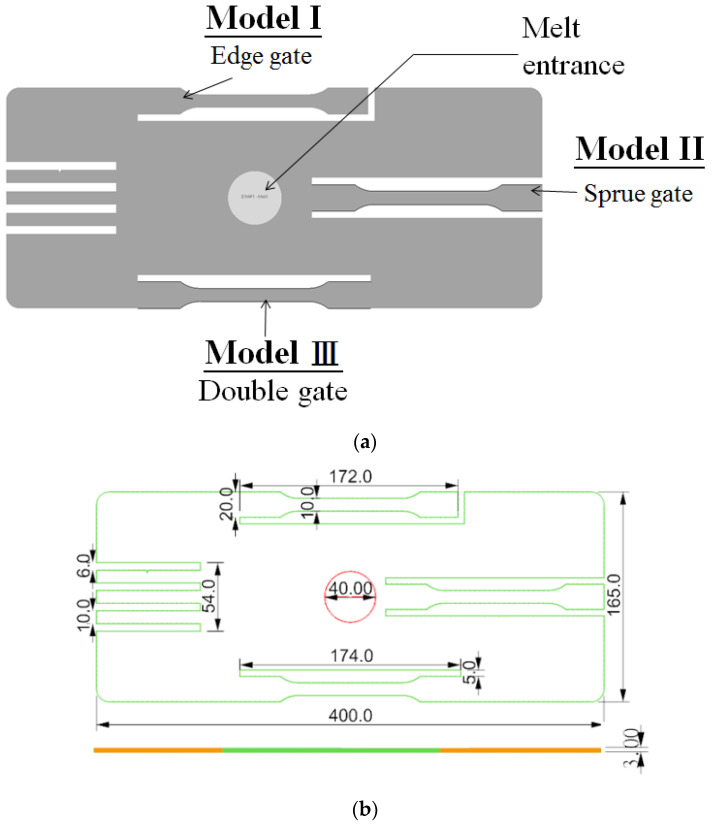
(**a**) The geometrical structure with three ASTM D638 specimens of different gate designs; (**b**) the dimension of the full sample.

**Figure 3 polymers-12-02274-f003:**
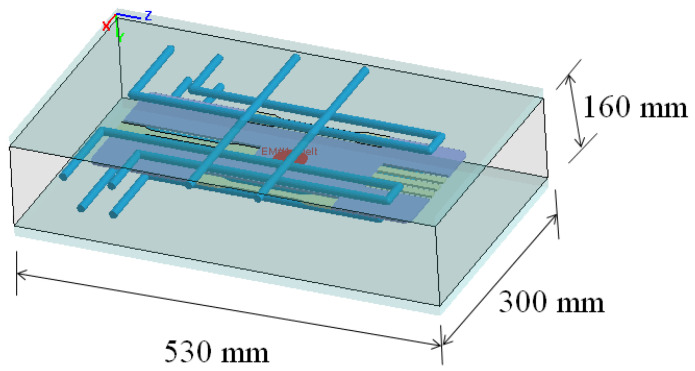
The moldbase and cooling channel layout.

**Figure 4 polymers-12-02274-f004:**
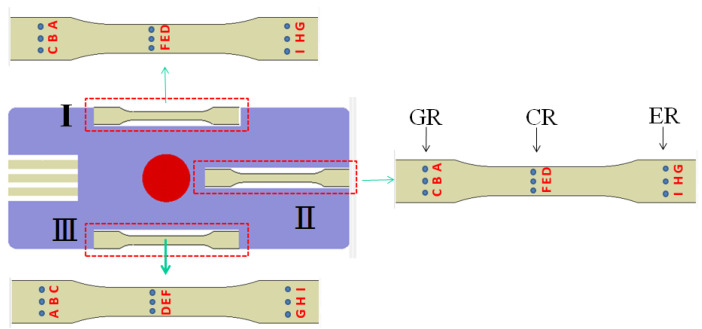
The observation locations at different specimen for FOD analysis.

**Figure 5 polymers-12-02274-f005:**
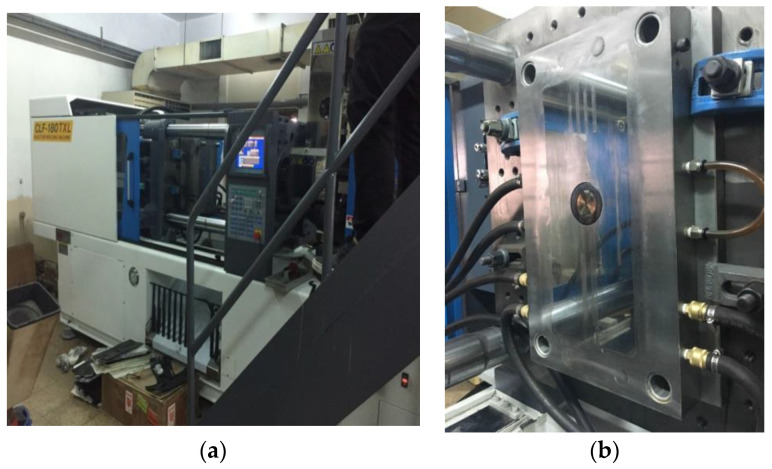
Machine and equipment setup: (**a**) Injection machine (Chuan Lih Fa Machinery Works Co., Model: CLF-180TXL), (**b**) the mold and cooling channel layout.

**Figure 6 polymers-12-02274-f006:**
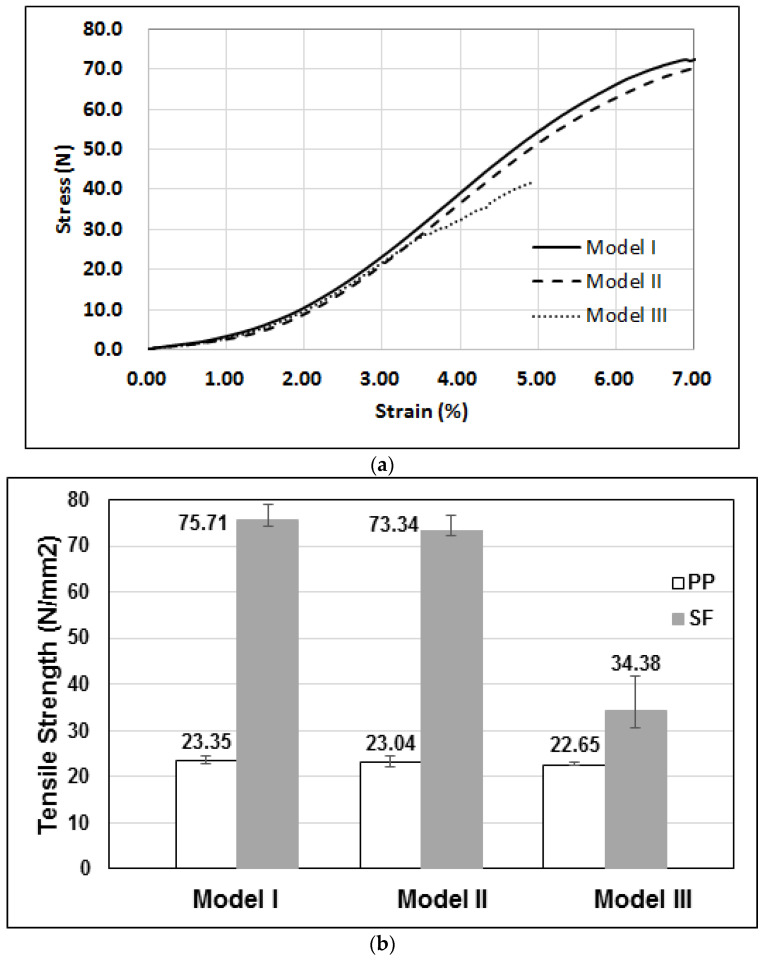
The mechanical properties of the PP-SF material with three different models, (**a**) the stress–strain relation, (**b**) the tensile strength properties.

**Figure 7 polymers-12-02274-f007:**
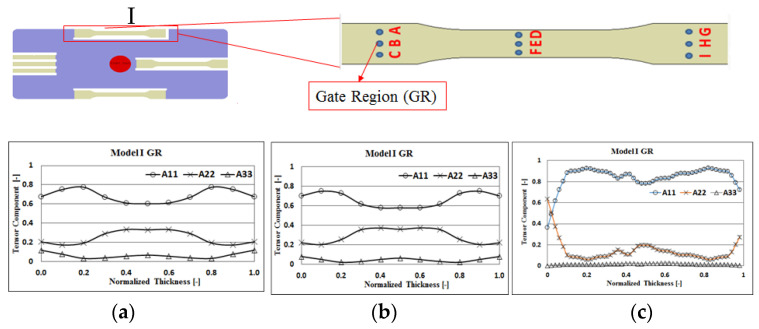
The comparison of the fiber orientation tensor components for Model I at Point B at GR: (**a**) simulation without flow–fiber coupling, (**b**) simulation with flow–fiber coupling, (**c**) experimental observation.

**Figure 8 polymers-12-02274-f008:**
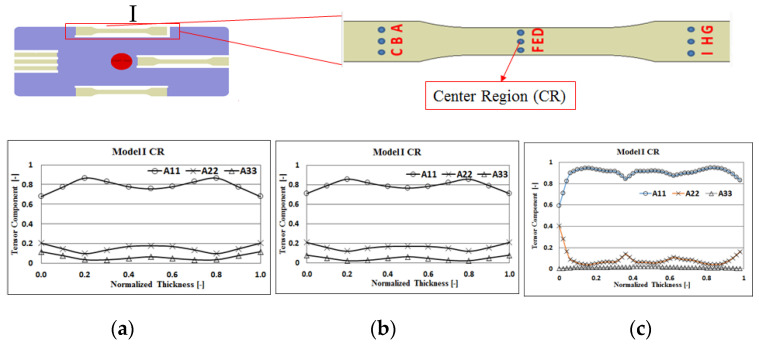
The comparison of the fiber orientation tensor components for Model I at Point E at CR: (**a**) simulation without flow–fiber coupling, (**b**) simulation with flow–fiber coupling, (**c**) experimental observation.

**Figure 9 polymers-12-02274-f009:**
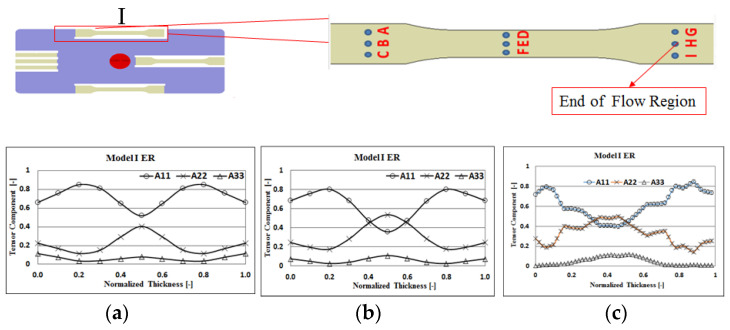
The comparison of the fiber orientation tensor components for Model I at Point H at ER: (**a**) simulation without flow–fiber coupling, (**b**) simulation with flow–fiber coupling, (**c**) experimental observation.

**Figure 10 polymers-12-02274-f010:**
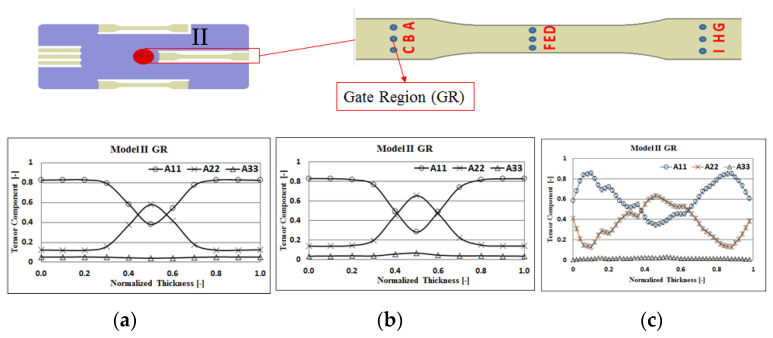
The comparison of the fiber orientation tensor components for Model II at Point B at GR: (**a**) simulation without flow–fiber coupling, (**b**) simulation with flow–fiber coupling, (**c**) experimental observation.

**Figure 11 polymers-12-02274-f011:**
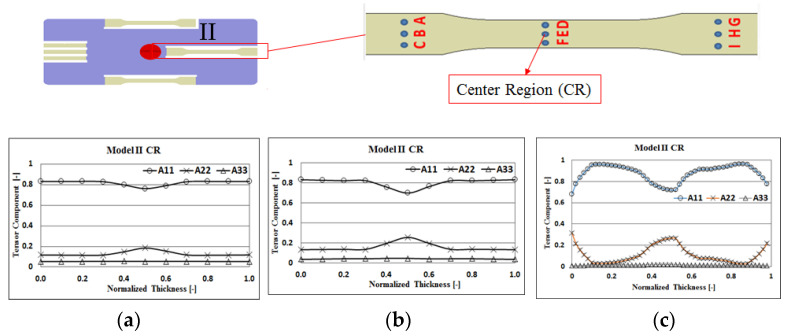
The comparison of the fiber orientation tensor components for Model II at Point E at CR: (**a**) simulation without flow–fiber coupling, (**b**) simulation with flow–fiber coupling, (**c**) experimental observation.

**Figure 12 polymers-12-02274-f012:**
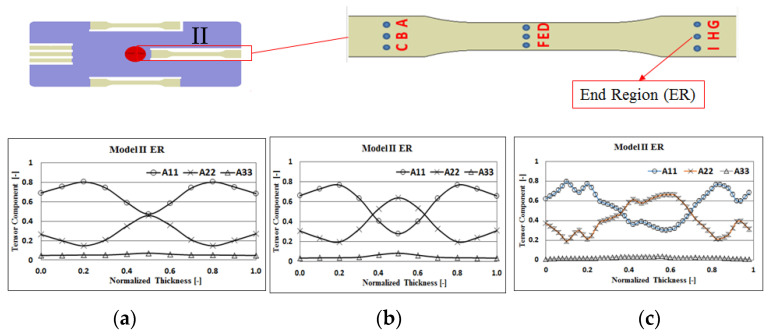
The comparison of the fiber orientation tensor components for Model II at Point H at ER: (**a**) simulation without flow–fiber coupling, (**b**) simulation with flow–fiber coupling, (**c**) experimental observation.

**Figure 13 polymers-12-02274-f013:**
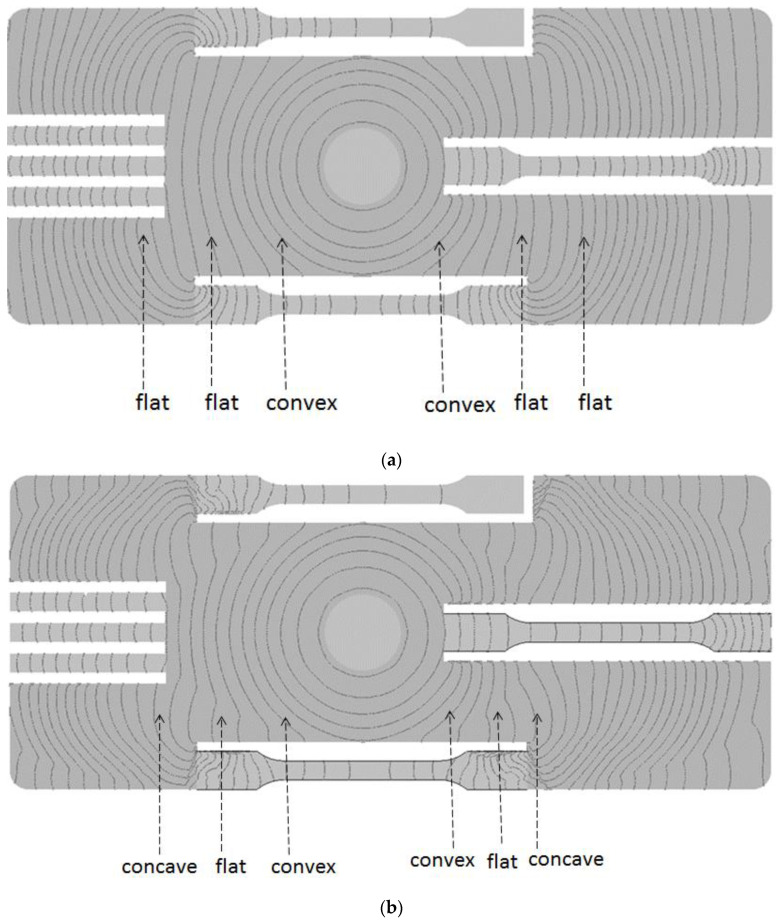
The melt flow front for the PP + SF composite in the injection molding simulation, (**a**) without flow–fiber coupling, (**b**) with flow–fiber coupling.

**Table 1 polymers-12-02274-t001:** Process condition setting for injection molding simulation.

Item	Process Condition
Filling time (s)	1.49
Packing time (s)	5
Packing pressure (MPa)	69.1
Cooling time (s)	15
Melt temperature (°C)	260
Mold temperature (°C)	25

**Table 2 polymers-12-02274-t002:** The parameters for iARD–RPR model in injection molding simulation.

Parameters	Values
C_I_	0.005
C_M_	0.5
α	0.7

**Table 3 polymers-12-02274-t003:** The parameters for the revised IISO model in flow–fiber coupling calculation.

Parameters	Values
$R_{T}^{0}$	2000
${\dot{\gamma }}_{C}$	10

**Table 4 polymers-12-02274-t004:** The tensile strength (N/mm^2^) of three models for each test.

Experiment	Model I	Model II	Model III
1	78.95	76.71	41.69
2	75.93	72.13	32.70
3	74.67	72.47	30.63
4	74.17	72.40	32.07
5	74.83	73.00	34.8
Average	75.71	73.34	34.38
Maximum	78.95	76.71	41.69
Minimum	74.17	72.13	30.63
Upper Error	3.24	3.37	7.31
Lower Error	1.54	1.21	3.74

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
