# Peer review of "Investigation on the Coupling Effects between Flow and Fibers on Fiber-Reinforced Plastic (FRP) Injection Parts"

_polymers, 2020, doi:10.3390/polym12102274_

Round 1
Reviewer 1 Report
The article is valuable and interesting. I have no comments and suggestions for the authors.
Author Response
Thank you very much for your comment.
Chao-Tsai Huang
Reviewer 2 Report
Title: Investigation on the coupling effects between flow and fibers on fiber-reinforced plastic (FRP) injection parts
Authors: Chao-Tsai Huang and Cheng-Hong Lai
Overall assessment:
The authors seem to examine the flow-fiber coupling effect on FRP injection parts using a geometry system with three ASTM D638 specimens by the numerical and experimental approaches. Such a topic may be interesting for several researchers in the literature. Even if so, however, the manuscript contains many casualties, ambiguities, and irrelevancies. therefore, I think that this version does not satisfy the criterion for the publication at all. If the authors are eager for the publication, they must conduct a thorough revision before the resubmission.
Followings are the specific comments that may be helpful for the resubmission:
Specific comments:
- The introduction section is lengthy and tedious to read. I strongly recommend the authors to compress it while focusing the target of the study.
- It is often difficult for many readers to understand the issues included in Section 2 appropriately. I recommend the authors to illustrate some figures to enhance the understanding of the readers.
- It is difficult to understand what the red circle and purple rectangle represent in Fig. 1a.
- PP and SF must not abbreviate at their first appearances.
- The descriptions on the materials used for the experiment are extremely terse.
- There are no descriptions on the method how to measure the strain in the tensile test; therefore, it is impossible to examine the validity of the stress-strain relations illustrated in Fig. 5a.
- It is difficult to understand how to average the stress-strain relations obtained from one testing condition because any detailed explanations are completely missing. Additionally, it is highly questionable that the decreasing tendency in Model I is really obtained by averaging the multiple testing data.
- In the line 223, the authors describe “To realize why the mechanical properties of the Model I are stronger than that of Model II”. When seeing Fig. 5, however, the mechanical properties of these models seem to be similar with each other.
- Although the A11, A22, and A33 values may be important for this study, the definitions of these values are not derived clearly. In particular, I cannot find any definition of the A33 value.
Recommendation
- I cannot continue the reviewing process in the present status. I think that the manuscript is indeed self-complacent; therefore, many readers cannot appreciate the essence of the study at all. When considering the resubmission, the authors must elaborate the manuscript thoroughly by conducting the careful and objective readings repeatedly.
Author Response
Please see the attachment.
Thank you very much for your help. Appreciate.
Chao-Tsai Huang

Reviewer 3 Report
The authors cover an interesting topic in the more applied field. There is for sure novelty by the presentation needs to be improved. Below my main comments:
Why the Cross model?
Table 1-3? Why these parameters. More context is needed.
Results: how well is the skin-layer covered?
The introduction is very high quality bit the discussion itself is too many figures. Specifically at the end. There is no embedding. I understand what has been done but I know these type of simulations. For a general reader a rewriting is needed, perhaps working with SI.
There are no major flaws in the work but I lack a bit through linkage of experiment and model. Now it is too much a technical report.
Author Response
Please see the attachment.
Thank you very much for your suggestions.
Chao-Tsai Huang

Round 2
Reviewer 2 Report
The revisions are adequately conducted; therefore I can recommend the revised version to be published as it is.
Reviewer 3 Report
The authors have improved the manuscript. I am specifically satisfied with the reply letter.